# Microbiome and Genetic Factors in the Pathogenesis of Liver Diseases

Dimitrina Miteva [1,2], Monika Peshevska-Sekulovska [2,3], Violeta Snegarova [4], Milena Peruhova [5], Georgi H. Vasilev [2,6], Georgi V. Vasilev [2,7], Metodija Sekulovski [2,8], Snezhina Lazova [2,9,10], Milena Gulinac [2,11], Latchezar Tomov [2,12], Antoaneta Mihova [13] and Tsvetelina Velikova [2,*]

1 Department of Genetics, Faculty of Biology, Sofia University St. Kliment Ohridski, 8 Dragan Tzankov Str., 1164 Sofia, Bulgaria; d.georgieva@biofac.uni-sofia.bg
2 Medical Faculty, Sofia University St. Kliment Ohridski, 1 Kozyak Str., 1407 Sofia, Bulgaria; mpesevska93@gmail.com (M.P.-S.); drgeorgivasilev@gmail.com (G.H.V.); vvasilev.georgi@gmail.com (G.V.V.); metodija.sekulovski@gmail.com (M.S.); snejina@lazova.com (S.L.); mgulinac@hotmail.com (M.G.); lptomov@nbu.bg (L.T.)
3 Department of Gastroenterology, University Hospital Lozenetz, Kozyak 1 Str., 1407 Sofia, Bulgaria
4 Clinic of Internal Diseases, Naval Hospital—Varna, Military Medical Academy, Medical Faculty, Medical University, Blvd. Hristo Smirnenski 3, 9000 Varna, Bulgaria; violetasnegarova@gmail.com
5 Department of Gastroenterology, Heart and Brain Hospital, Zdrave 1 Str., 8000 Burgas, Bulgaria; mperuhova@gmail.com
6 Laboratory of Hematopathology and Immunology, National Specialized Hospital for Active Treatment of Hematological Diseases, "Plovdivsko Pole" Str. 6, 1756 Sofia, Bulgaria
7 Department of Emergency Medicine and Clinic of Neurology, University Hospital "Sv. Georgi", Blvd. Peshtersko Shose 66, 4000 Plovdiv, Bulgaria
8 Department of Anesthesiology and Intensive Care, University Hospital Lozenetz, 1 Kozyak Str., 1407 Sofia, Bulgaria
9 Pediatric Department, University Hospital "N. I. Pirogov", 21 "General Eduard I. Totleben" Blvd, 1606 Sofia, Bulgaria
10 Department of Healthcare, Faculty of Public Health, "Prof. Tsekomir Vodenicharov, MD, DSc", Medical University of Sofia, Bialo More 8 Str., 1527 Sofia, Bulgaria
11 Department of General and Clinical Pathology, Medical University of Plovdiv, Bul. Vasil Aprilov 15A, 4000 Plovdiv, Bulgaria
12 Department of Informatics, New Bulgarian University, Montevideo 21 Str., 1618 Sofia, Bulgaria
13 SMDL Ramus, Department of Immunology, Blvd. Kap. Spisarevski 26, 1527 Sofia, Bulgaria; toni02m@yahoo.com
* Correspondence: tsvelikova@medfac.mu-sofia.bg

**Abstract:** Our genetic background has not changed over the past century, but chronic diseases are on the rise globally. In addition to the genetic component, among the critical factors for many diseases are inhabitants of our intestines (gut microbiota) as a crucial environmental factor. Dysbiosis has been described in liver diseases with different etiologies like non-alcoholic fatty liver disease (NAFLD), alcohol-related liver disease (ALD), viral hepatitis, autoimmune hepatitis (AIH), primary sclerosing cholangitis (PSC), primary biliary cholangitis (PBC), cirrhosis, hepatocellular carcinoma (HCC). On the other hand, new technologies have increased our understanding of liver disease genetics and treatment options. Genome-wide association studies (GWAS) identify unknown genetic risk factors, positional cloning of unknown genes associated with different diseases, gene tests for single nucleotide variations (SNVs), and next-generation sequencing (NGS) of selected genes or the complete genome. NGS also allowed studying the microbiome and its role in various liver diseases has begun. These genes have proven their effect on microbiome composition in host genome–microbiome association studies. We focus on altering the intestinal microbiota, and supplementing some bacterial metabolites could be considered a potential therapeutic strategy. The literature data promote probiotics/synbiotics role in reducing proinflammatory cytokines such as TNF-$\alpha$ and the interleukins (IL-1, IL-6, IL-8), therefore improving transaminase levels, hepatic steatosis, and NAFLD activity score. However, even though microbial therapy appears to be risk-free, evaluating side effects related to probiotics or synbiotics is imperative. In addition, safety profiles for long-term usage should be researched. Thus, this review focuses on the human microbiome and liver diseases, recent

GWASs on liver disease, the gut-liver axis, and the associations with the microbiome and microbiome during/after liver disease therapy.

**Keywords:** genomics; liver disease; microbiome; gut microbiota; NAFLD; liver cirrhosis; autoimmune liver disease

## 1. Introduction

Over the past century, our genetic background has not changed, but chronic diseases are on the rise globally. In addition to the genetic component, the critical factors for many diseases are lifestyle, eating changes, exposure to drugs, xenobiotics, alcohol, smoking, polluted air, etc. [1]. The role of the inhabitants of our intestines (gut microbiota) is also seen as a critical environmental factor. To date, it is considered that there is a direct connection between gut dysbiosis and chronic diseases. It has been found that humans are composed of trillions of cells, about $3 \times 10^{13}$ eukaryotic cells, and the microbiome is about $4 \times 10^{13}$ colonizing microbes, i.e., the ratio is very close to 1:1 [1].

Alterations in the microbiota can decrease microbial diversity and increase proinflammatory species. An imbalance of the normal gut microbiota has been linked with gastrointestinal inflammatory conditions, autoimmune diseases, chronic liver diseases, hormonal and metabolism disorders, and neuropsychiatric manifestations [2]. These observations are presented in Figure 1.

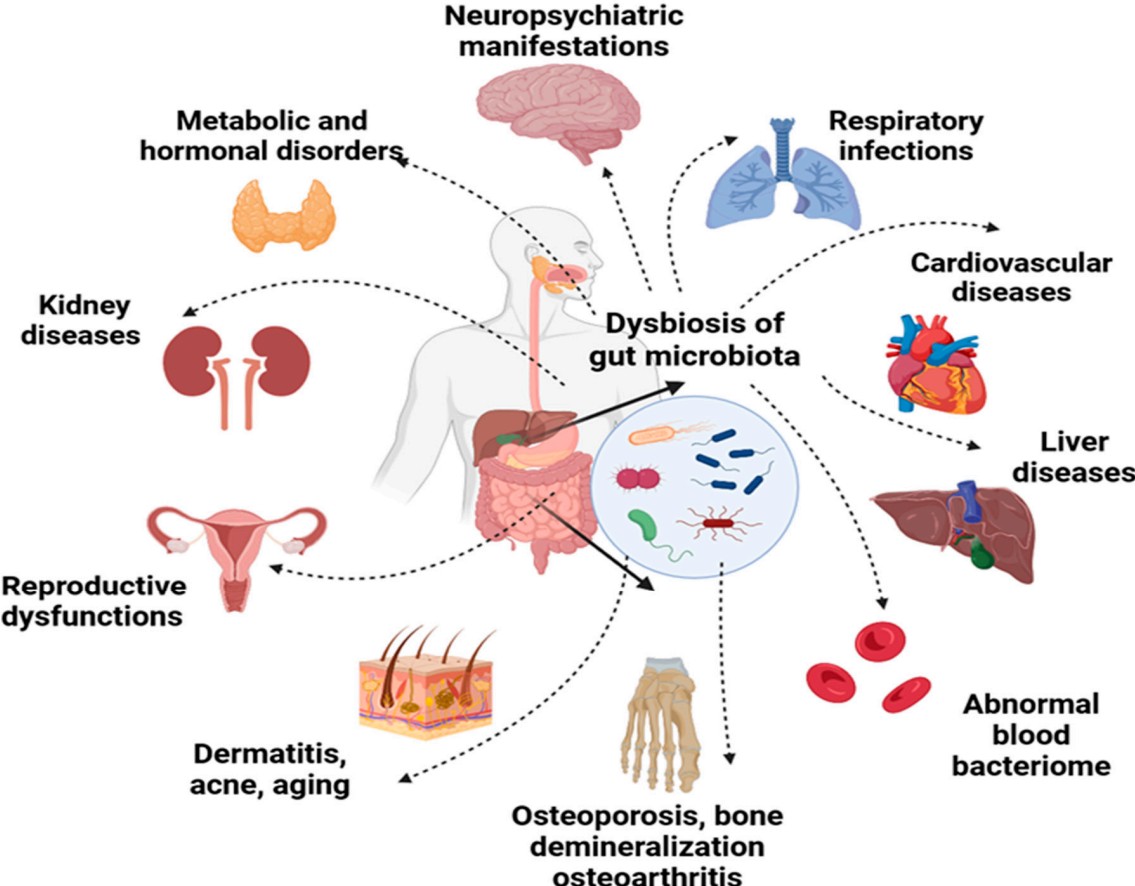

**Figure 1.** Adverse clinical outcomes in the case of gut microbial dysbiosis. Microbial communities are in symbiosis with the host, and dysbiosis can lead to the dysregulation of various functions and diseases, including liver diseases, cardiovascular diseases, metabolic and respiratory diseases, cancer, etc.

The pathology of liver disorders is due to connections between genetic and environmental factors (in particular, the microbiome). The next-generation sequencing (NGS) made it possible to study the microbiome, and its role in various liver diseases has begun. Using different animal models, the gut microbiome in liver diseases has been studied (fibrosis, cirrhosis, alcoholic-related liver disease, cancer, etc.) [3–5].

The interplay between the liver and the gut is bidirectional. The liver secretes primary bile acids (BAs) and antimicrobial molecules (angiogenin and IgA) into the biliary tract. These molecules enter the lumen and help maintain microbial balance within the body. Microbial metabolites like sBAs, microbial- (or pathogens-) associated molecular patterns (MAMPs/PAMPs), trimethylamine (TMA), trimethylamine N-oxide (TMAO), etc., pass via the portal vein into the liver and affect its correct functioning [6,7]. This continuous recirculation of molecules through the blood capillaries can strongly affect the intestinal barrier and alter the gut-liver axis. The intestinal mucosal barrier is the functional structure where the interactions between the gut and the liver occur, limiting the spread of microbes, viruses, and toxins but allowing the nutrients to reach the circulation and the liver (Figure 2) [8].

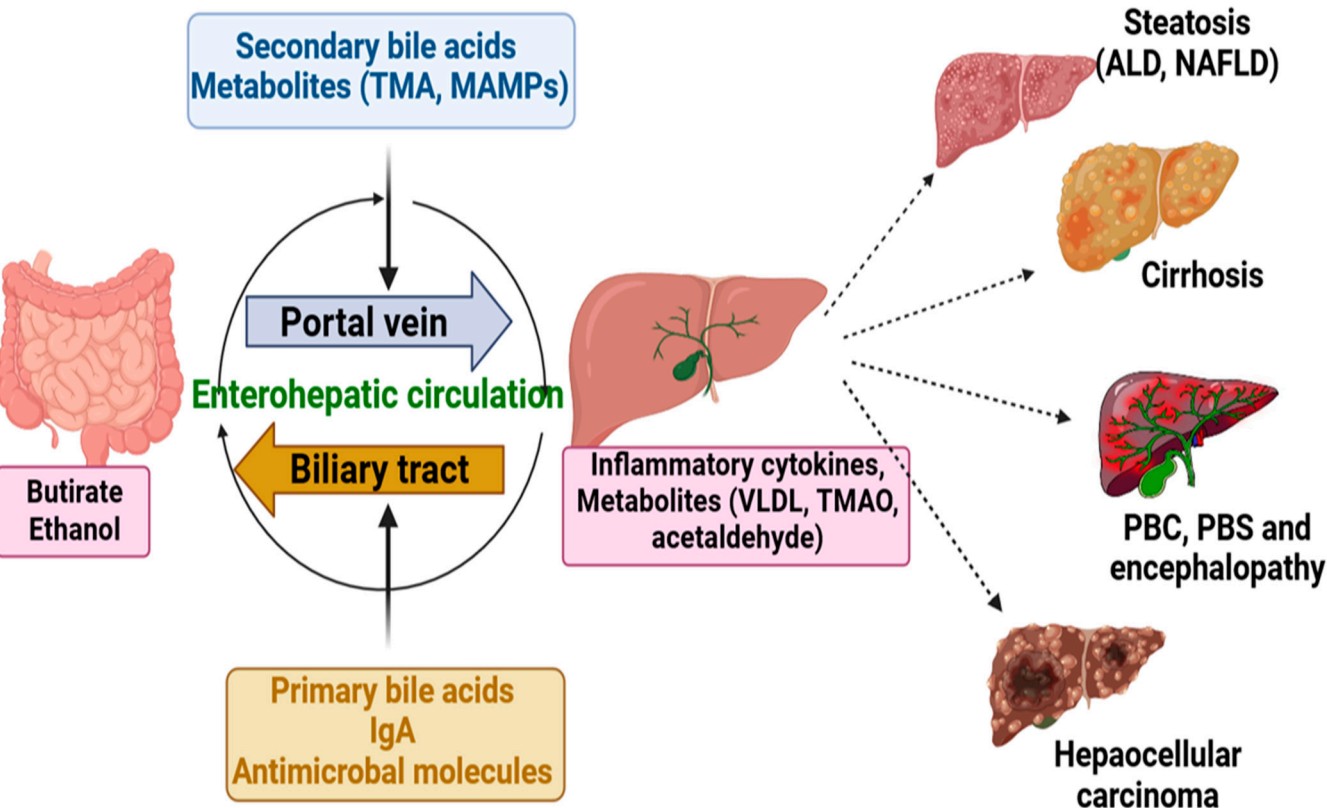

**Figure 2.** Gut-liver axis: bidirectional communication and the physiological manifestations of liver injury. (TMA—Trimethylamine; TMAO—Trimethylamine N-oxide; MAMPs—Microbial-associated molecular patterns; BAs—bile acids; VLDL—Very low-density lipoprotein; ALD—alcohol-related liver disease; NAFLD—non-alcoholic fatty liver disease; PBC—primary biliary cholangitis; PSC—primary sclerosing cholangitis).

Over the past decade, scientific evidence for interactions between the microbiota and host genes and gene expression has been sparse [9]. There is still no clear answer as to whether changes in the microbiota lead to a disease or are the cause of a disease. However, it has been established that the microbiome is causal in various diseases (metabolic, gastroenterological, and liver diseases, cardiovascular, allergies, and neurological disorders). That dysbiosis leads to adverse clinical outcomes [3,10–13].

When we study the genes of the microbiome and the host, we can gain much useful information about the interactions between them (Figure 3). The human genome contains many protein-coding genes that are regulated by host-specific factors and environmental signals. The microbial genomes in each of our microbiomes, sometimes called our "second genome", also contain many genes and expand the coding potential of our own genome [14]. The mutation profile of disease genes varies. Conducting GWAS is necessary to identify numerous genes/genetic variants associated with a particular phenotype or risk of diseases worldwide. Therefore, genetics is helping us to improve our understanding of the pathophysiology of liver diseases. Suppose we can understand which human genes have a significant association and/or predispositions with liver diseases. In that case, we will be able to offer precise diagnosis and personalized treatment for patients.

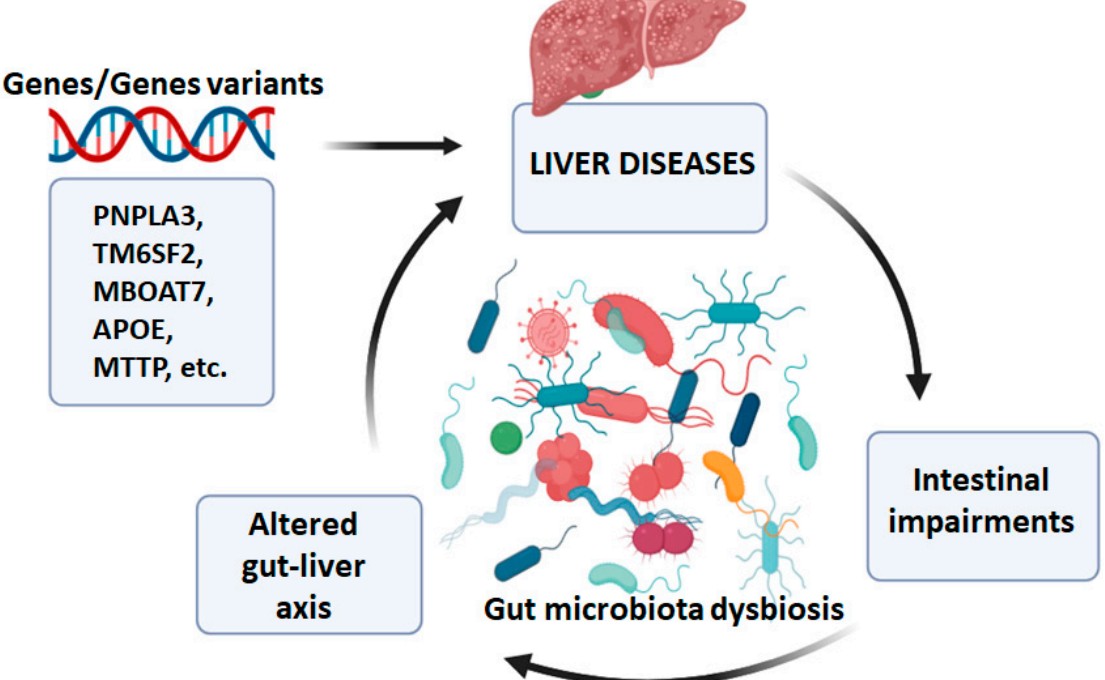

**Figure 3.** Mutual effect among genes/genes variants and microbiota in liver diseases. Different factors alter gut metabolism and barrier function, contributing to gut microbiota dysbiosis. Disbalance in the composition of microbiota, in turn, leads to altered gut-liver axis and the physiological manifestations of liver injury. The interaction between microbiota and host genetic variants also plays a crucial role in the complex pathogenesis of liver diseases. The mechanisms by which these factors interplay with each other are not fully understood, highlighting the need for future studies to clarify genetic and environmental factors affecting the microbial composition and its relationship with human diseases.

With the development of Omics-technologies, more and more microbial metabolites and their interaction with immunity help to unravel liver pathogenesis. This review focuses on recent genome-wide association studies (GWASs) on liver disease, the gut-liver axis, and the associations with the microbiome and microbiome during/after liver disease therapy. A better understanding of the contribution of gut microbes to liver diseases may help us with new treatments.

Different liver diseases follow similar pathophysiological mechanisms in which the liver, after an injury, regenerates. It has been established that the intestinal microbiota also dynamically changes and can lead to these processes. Gut dysbiosis has been found in liver diseases with different etiologies like non-alcoholic fatty liver disease (NAFLD), alcohol-related liver disease (ALD), viral hepatitis, autoimmune hepatitis (AIH), primary sclerosing cholangitis (PSC), primary biliary cholangitis (PBC), cirrhosis, hepatocellular carcinoma [15–18].

## 2. Human Microbiome and Liver Diseases

Recent studies focused on gut microbiota's role in the pathogenesis and treatment of liver diseases. Although the causal relationship between the role of microbiota and liver disease is not fully known, several studies have examined the effectiveness of fecal microbiota transplantation (FMT) and the impact of pre-pro- and synbiotics in liver disease [18].

A study including 58 adults diagnosed with NAFDL, divided into two groups, one receiving a Multi-probiotic product and the other a placebo, found reductions of aspartate transferase (AST), gamma-glutamyl transpeptidase (GGT), tumor necrosis factor (TNF)-a and IL-6 [19].

Another study, again in patients suffering from NAFDL, found that after administration of synbiotics, there was a significant decrease in AST, total cholesterol, triacylglycerol, and steatosis (based on Fibro scan) [20].

Various studies have been conducted on patients diagnosed with ALD. For example, administering *Bifidobacterium bifidum* and *Lactiplantibacillus plantarum* for 5 days significantly reduced liver transaminases AST and alanine transaminase (ALT) [21]. Another study, in patients diagnosed with ALD and cirrhosis, showed a decrease in TNF-a, an increase in albumin levels, and a stabilization of lipopolysaccharide (LPS) levels after administering *Lactobacillus subtilis* and *Streptococcus faecium* (daily for 7 days) [22].

A double-blind, randomized clinical trial (RCT) (n = 39) study in patients with cirrhosis shows Improvement of liver function/Child-Pugh score and lowering of endotoxemia after administration of *Escherichia coli Nissle* for 42 days [23]. Applying various pre-, pro-, and synbiotics improves laboratory parameters and liver parameters in patients with NAFLD, ALD, cirrhosis, or hepatocellular carcinoma [18].

Since FMT has been used with great success in treating antibiotic-resistant *Clostridium difficile*, numerous studies have been conducted on the therapeutic possibilities of FMT, including its use in liver diseases [24]. A pilot study in patients with NAFDL showed significantly reduced insulin resistance associated with changes in intestinal microbiota [25]. A new study on the role of FMT in patients diagnosed with cirrhosis found that those who received FMT had reduced hospitalization rates and improved cognition and dysbiosis. In addition, 5 months after the procedure, no patient in the FMT group developed hepatic encephalopathy compared to the control group [26].

### 2.1. Autoimmune Hepatitis (AIH) and Gut Microbiome

Autoimmune hepatitis (incidence 0.9–2/100,000 populations per year) is chronic liver inflammation and occurs when your body's immune system attacks liver cells. Gut microbial dysbiosis, characterized by a reduced abundance of beneficial bacteria, is often found in patients with AIH. Antibodies against soluble liver antigen/liver–pancreas (anti-SLA/LP) are a specific serological marker for autoimmune hepatitis (AIH). They are associated with a more severe form of the disease or worse survival. Up to 10–20% of people diagnosed with AIH have these specific autoantibodies to SLA/LP [27,28].

There is believed to be a molecular mimicry between soluble liver antigen/liver–pancreas (SLA/LP) and bacterial surface antigen [26]. Such a study was presented by Paiardini et al., who found a structural similarity between a section of the surface antigen PS 120 from Rickettsia spp. and immunodominant regions of the SLA/LP autoepitope [29]. This finding lends weight to the concept that molecular mimicry might cause AIH [26]. Therefore, it is not surprising that the molecular mimicry between human PDC-E2 and *E. coli* PDC-E2 and infection with *E. coli* led to the generation of disease-specific antimitochondrial autoantibodies [30].

Ngu et al. demonstrated in their study, including 72 AIH patients and 144 healthy controls, that antibiotic exposure in AIH patients within 12 months before the diagnosis establishment was an independent risk factor for AIH manifestation [31]. In addition, literature data showed that probiotic intake could alleviate gut dysbiosis [32,33].

Liu et al. investigated the effects of compound probiotics in the AIH mouse model, which were also injected with dexamethasone intraperitoneally for 42 days [34]. According

to their findings, these therapies successfully reduced the number of inflammatory cells in the liver, serum transaminase levels, and both Th1 and Th17 cells. However, the only group that showed an increase in Treg cells was the probiotic group, which suggests that it has an immunomodulatory effect. Combined probiotics may increase ileal barrier function and boost intestinal flora diversity [35]. They reduce the transport of gut-derived LPS to the liver, therefore inhibiting the activation of the TLR4/NF-B pathway. Consequently, the synthesis of proinflammatory factors is inhibited, making AIH remission easier to achieve [34,35].

Another substantial study has confirmed the microbiota variation between AIH patients and healthy controls [36]. They compared gut microbiota using fecal gene sequencing and established that the microbial communities in both groups differed significantly. In the AIH group, the abundance of *Verrucomicrobiota* increased considerably, whereas the abundance of *Lentisphaerae* and *Synergistetes* was significantly decreased. Furthermore, 15 genera, including *Veillonella*, *Faecalibacterium*, and *Akkermansia*, were enriched in the AIH patients compared to the HCs, whereas 19 genera, including *Pseudobutyrivibrio*, *Lachnospira*, and *Ruminococcaceae*, decreased.

Wei et al. have also conducted a cross-sectional study about gut microbiome alteration of AIH and healthy controls [37]. They performed 16S rRNA sequencing before corticosteroid therapy and found a depletion of obligate anaerobes and an expansion of potential pathobionts such as *Veillonella*. Moreover, the authors reported a strong correlation between *Veillonella dispar* abundance and serum level of aspartate aminotransferase (AST) and liver inflammation.

Consistent with this report, Liwinski et al. supported the facts about altered overall microbiota composition, reduced biodiversity, and relative amount of beneficial anaerobic species [38]. One of the most interesting findings made by this research was that a considerable decrease in Bifidobacterium failed to achieve remission of liver inflammation in AIH patients. Importantly, their research has demonstrated that the abnormalities in gut microbiota in AIH are unique to the disease. AIH and primary PBC can be differentiated from one another quite effectively based on the microbiota profile. Hence, the gut microbiome's functional changes in AIH can be used as non-invasive biomarkers to assess disease activity.

## 2.2. Primary Biliary Cholangitis (PBC) and Gut Microbiome

Regarding the correlation between PBC (incidence 1.76/100,000 person-years) and gut microbiome, Li et al. conducted a study of gut microbiome diversity in 42 patients with early-stage PBC [39]. They established that patients with PBC had lower levels of some potentially beneficial gut bacteria, such as *Acidobacteria*, *Lachnobacterium* sp., *Bacteroides eggerthii*, and *Ruminococcus bromii*. However, these patients also had higher levels of some bacterial taxa that contained opportunistic pathogens, such as—*Proteobacteria*, *Enterobacteriaceae*, *Neisseriaceae*, *Spirochaetaceae*, *Veillonella*, *Streptococcus*, *Klebsiella*, *Actinobacillus pleuropneumoniae*, *Anaeroglobus geminatus*, *Enterobacter asburiae*, *Haemophilus parainfluenzae*, etc. [39]. Therefore, the altered gut microbiota might be the hidden villain behind PBC onset.

Tang et al. studied the microbiome diversity in ursodeoxycholic acid (UDCA)-naïve PBC patients. The authors established an overabundance of *Haemophilus*, *Klebsiella*, *Streptococcus*, and *Veillonella* in these cohorts compared to healthy controls. They also reported that after six months of treatment with UDCA, the altered amount of six genera linked to PBC was restored [40].

Furukawa et al. studied not only gut disbalance in the composition of microbiota in PBC patients but also different clinical profiles and biochemical responses to one year of UDCA therapy [41]. Furthermore, among the patients who did not respond to UDCA treatment, a lower amount of *Faecalibacterium* was detected, which led to the conclusion that this isolate could predict PBC prognosis.

A recent study investigated fecal microbiota and metabolic profiles in PBC patients [42]. In advanced fibrosis, both fecal acetate and short-chain fatty acids (SCFA) were shown to be elevated profiles. The microbiota of advanced fibrosis patients displayed lower levels of alpha diversity, higher levels of *Weisella*, and distinctive bacterial composition.

## 2.3. Primary Sclerosing Cholangitis (PSC) and Gut Microbiome

PSC is an uncommon, chronic liver disease (incidence 0.5–1.3/100,000 person-years) in which inflamed internal and external bile ducts in the liver are narrowed or blocked. The bile causes liver damage. The pancreas (perinuclear Anti-Neutrophil Cytoplasmic Antibodies) are critical players in PSC pathogenesis and their cross-reaction with bacterial proteins, which predispose to an abnormal immune response to the gut microbiome [43].

A recent study used metagenomic shotgun sequencing on a cohort from Germany and Norway. This study demonstrated that there was a decrease in the diversity of microbial genes in PSC, as well as an increase in the predominance of species belonging to the genus *Clostridium* and depletion of species belonging to the genus *Eubacterium* and *Ruminococcus obeum* [44]. PSC patients have discernible variations in the number of genes associated with branched-chain amino acid synthesis and vitamin B6 synthesis. Although the prevalence of *Veillonella* was lower than in earlier 16S-based investigations, the authors observed an elevated prevalence of certain *Veillonella* species in patients with PSC. This work demonstrates the benefits of moving to metagenomic shotgun sequencing and adding blood and stool samples into the analysis.

Iwasawa et al. established the reduction of butyrate-producing anaerobes in patients with PSC pediatric onset [45]. Additionally, it is essential to underline that PSC is distinguished from inflammatory bowel disease (IBD) populations by a particular form of dysbiosis. There seems to be no difference between having PSC solely and having PSC-IBD, which suggests that liver pathology is the primary consequence of microbial dysbiosis [45].

However, it is not entirely correct to blame *Veillonella* only for PSC. As mentioned above, *Veillonella*'s increased amount is registered in cirrhosis of various origins, such as AIH, PBC, or non-hepatic disorders, such as treatment-naïve Crohn's disease [46,47].

## 2.4. Alcohol-Related Liver Disease (ALD) and Gut Microbiota

Among the key triggering factors for end-stage liver disease is alcohol consumption. Special attention has been paid to the alcohol effect on the gut mucosa. It has been reported that the intestinal mucosal barrier could be injured by ethanol and its oxidative and non-oxidative metabolites in conjunction with inflammation brought on by intestinal dysbiosis [48]. Alterations in the levels of several microbial metabolites, including amino acids, bile acids, and short-chain fatty acids, are intimately related to gut dysbiosis in ALD (prevalence of 4.8% worldwide). Intestinal barrier-related proteins can be further influenced by alcohol-caused dysbiosis. These proteins include mucin 2, bile acid-related receptors, and aryl hydrocarbon receptor (AhR). Abnormal changes in these proteins also contribute to intestinal mucosal barrier injury and hepatic steatosis. Because of the damage to the intestinal barrier, bacteria, and fungi that originate in the gut, as well as their toxins, including LPS and beta-glucan, can reach portal circulation and, consequently, liver parenchyma and contribute to the advancement of fibrosis and inflammation associated with ALD [48].

Some microorganisms are responsible for developing ALD, while others have positive benefits and even protective effects. Yan et al. found out that mice on an alcohol diet demonstrated an abundance of *Bacteroidetes* and *Verrucomicrobia* compared to mice on a control diet [49]. In addition, Kirpich et al. have shown that in contrast to the healthy group, the numbers of *Bifidobacteria*, *Lactobacilli*, and *Enterococci* are dramatically reduced in the alcoholics [21]. Another study reveals that in comparing alcohol-induced cirrhotic patients to healthy people, the proportion of Bacteroidetes was much lower in the cirrhosis group. In contrast, the proportion of *Proteobacteria* and *Fusobacteria* was significantly higher in the latter group [50].

Tuomisto et al. described that individuals with alcoholic cirrhosis had 27 times more *Enterobactericaea* in their stools than healthy volunteers [51]. These findings prove that gut microbiota plays a crucial role in cirrhosis patients.

### 2.5. Non-Alcoholic Fatty Liver Disease (NAFLD) and Gut Microbiome

Non-alcoholic fatty liver disease (NAFLD) is a complex systemic disease characterized by hepatic lipid buildup, lipotoxicity, insulin resistance, gut dysbiosis, and inflammation, with a worldwide prevalence of 47 per 1000 population [52]. However, NAFLD will be better managed by researchers and clinicians if they understand that the disease results from a complex interaction between metabolism, gut microbiome, and the immune response.

Recently, the literature data have highlighted the role of gut dysbiosis and its notorious consequences, such as increased free fatty acid absorption, bacterial migration, and release of toxic bacterial products, lipopolysaccharide (LPS), and proinflammatory cytokines that initiate and sustain inflammation [16,52]. These studies established that adult NAFLD patients have different patterns of gut dysbiosis in contrast to pediatric patients. An Overabundance of *Proteobacteria*, *Enterobacteriaceae*, and *Escherichia* spp., with depletion in *Faecalibacterium prausnitzii* and *Akkermansia muciniphila*, was registered in adult patients. In contrast, the young showed reduced *Oscillospira* spp. and abundant levels of *Dorea*, *Blautia*, *Prevotella copri*, and *Ruminococcus* spp.

Li et al. underlined the association of gut dysbiosis with bile acid alterations and reduced butyrate production, considering that microbiome diversity could be at the base of NAFLD pathogenesis [53]. Behary et al. have discovered that individuals with NAFLD-cirrhosis have gut dysbiosis and that developing hepatocellular carcinoma (HCC) is associated with compositional and functional alterations in the microbiota [54]. According to their research, the gut microbiota of patients with NAFLD-associated HCC have a distinct microbiome/metabolomic profile and can affect the peripheral immune response.

In conclusion, altering the intestinal microbiota and supplementing some bacterial metabolites could be considered a potential therapeutic strategy. The literature data available promote probiotics/synbiotics role in reducing proinflammatory cytokines such as TNF-$\alpha$ and the interleukins (IL-1, IL-6, IL-8), therefore improving transaminase levels, hepatic steatosis, and NAFLD activity score. However, even though microbial therapy appears to be risk-free, evaluating side effects related to probiotics or synbiotics is imperative. In addition, safety profiles for long-term usage should be researched.

### 2.6. Liver Cirrhosis and Gut Microbiota

The potential treatments for liver cirrhosis (prevalence of 115.5/100,000 person-years) that modulate the gut microbiota and gut-liver axis have gained much interest lately. According to recent research, probiotics' gut microbiome modulation slows liver disease progression [55,56].

In addition to these changes, it has been demonstrated that alterations in bacterial function, such as increased endotoxin release and decreased conversion of primary bile acids to secondary bile acids, can lead to cirrhosis [57]. Moreover, dysbiosis could cause an increased intestinal permeability or so-called leaky gut, resulting in bacterial endotoxins passing to portal circulation and activating various inflammatory signaling pathways [58]. Bajaj et al. in their study found that the mucosal microbiota of cirrhotic, particularly patients with hepatic encephalopathy, differs significantly from that of healthy controls and an overgrowth of potentially pathogenic genera, both of which are associated with poor cognition and inflammation [59].

Another study by Chen et al. has shown an overabundance of *Veillonella*, *Megasphaera*, *Dialister*, *Atopobium*, and *Prevotella* in the gut microbiome of cirrhotic patients [55]. Consistent with the abovementioned data, an interesting theory has been published by Kakiyama et al. [60]. The authors have shown that rifaximin treatment in cirrhosis patients leads to reduced *Veillonellaceae* concentration and decreased secondary/primary bile acids (BAs) ratios. They established that cirrhotic patients had reduced conversion of pri-

mary to secondary BAs, leading to an overabundance of *Enterobacteriaceae*, *Lachnospiraceae*, *Ruminococcaceae*, and *Blautia*.

An overview of the studies on microbiome in autoimmune liver diseases is presented in Table 1 [38,46,61–85].

**Table 1.** Studies focused on associations between microbiome and autoimmune liver diseases.

| Ref. | N of Subjects | Liver Disease | Sample | Method | Enriched Taxa | Type Microorganism |
|---|---|---|---|---|---|---|
| Bode et al. [61] | 27 | ALD | Jejunal aspirate | Culture | Coliform microorganisms, Gram-negative anaerobic bacteria, endospore-forming rod | Bacteria |
| Mutlu et al. [62] | 19 | ALD | Mucosa | 16S rRNA | *Bacilli*, *Gammaproteobacteria* | Bacteria |
| Wang et al. [63] | 8 | ALD | Mucosa | 16S rRNA | Mucosa-assisted bacteria | Bacteria |
| Leclercq et al. [64] | 50 | ALD | Stool | 16S rRNA | At the family level: *Lachnospiraceae, Incertae sedis XIV* At the genus level: *Dorea, Blautia, Megasphaera* | Bacteria |
| Grander et al. [65] |  | ALD | Stool | 16S rRNA | *A. muciniphila* | Bacteria |
| Duan et al. [66] | 75 | ALD | Stool | 16S rRNA | *Veillonella, Escherichia/Shigella, Megasphaera* | Bacteria |
| Lang et al. [67] | 72 | ALD | Stool | 16S rRNA | *Veillonella, Enterococcus* | Bacteria |
| Yang et al. [68] | 20 | ALD | Stool | ITS | *Candida* | Fungi |
| Lang et al. [69] | 74 | ALD | Stool | ITS | *Candida* | Fungi |
| Chu et al. [70] | 133 | ALD | Stool | Culture + qPCR | *Candida* | Fungi |
| Jiang et al. [71] | 125 | ALD | Stool | Metagenomics | *Escherichia phage, Enterobacteria phage, Enterococcus phage, Parvoviridae, Herpesviridae* | Virus |
| Zhu et al. [72] | 47 | NASH/ NAFLD | Stool | 16S rRNA | *Proteobacteria, Enterobacteriaceae, Escherichia* | Bacteria |
| Mouzaki et al. [73] | 33 | NASH/ NAFLD | Stool | PCR | *Clostridium coccoides* | Bacteria |
| Alferink et al. [74] | 478 | NASH/ NAFLD | Stool | 16S rRNA | *Ruminococcus gauvreauiigroup, Ruminococcus gnavusgroup* | Bacteria |
| Loomba et al. [75] | 86 | NASH/ NAFLD | Stool | Metagenomics | *Proteobacteria, Escherichia coli* | Bacteria |
| Lang et al. [76] | 73 | NASH/ NAFLD | Stool | 16S rRNA + Metagenomics | *Escherichia phage, Enterobacteriaphage, Lactobacillus phage* | Virus |
| Wei et al. [37] | 91 | AIH | Stool | 16S rRNA | *Veillonella, Klebsiella, Streptococcus, Lactobacillus* | Bacteria |

**Table 1.** *Cont.*

| Ref. | N of Subjects | Liver Disease | Sample | Method | Enriched Taxa | Type Microorganism |
|---|---|---|---|---|---|---|
| Liwinski et al. [38] | 72 | AIH | Stool | 16S rRNA | *Veillonella, Klebsiella, Streptococcus* | Bacteria |
| Liwinski et al. [46] | 99 | PBC | Stool | 16S rRNA | *Veillonella, Klebsiella, Streptococcus* | Bacteria |
| Lv et al. [39] | 42 | PBC | Stool | 16S rRNA | *Proteobacteria, Enterobacteriaceae, Neisseriaceae, Spirochaetaceae, Veillonella, Streptococcus, Klebsiella, Actinobacillus, Anaeroglobus, Enterobacter, Haemophilus, Megasphaera, Paraprevotella* | Bacteria |
| Tang et al. [40] | 97 | PBC | Stool | 16S rRNA | *Haemophilus, Veillonella, Clostridium, Lactobacillus, Streptococcus, Pseudomonas, Klebsiella, Enterobacteriaceae* | Bacteria |
| Furukawa et al. [41] | 149 | PBC | Stool | 16S rRNA | *Lactobacillales* | Bacteria |
| Torres et al. [77] | 20 (19 with IBD) | PSC | Mucosa | 16S rRNA | *Barnesiellaceae, Blautia, Ruminococcus* | Bacteria |
| Quraishi et al. [78] | 11 | PSC—IBD | Mucosa | 16S rRNA | *Lachnospiraceae, Escherichia, Megasphera* | Bacteria |
| Pereira et al. [79] | 80 | PSC | Bile | 16S rRNA | *Streptococcus* | Bacteria |
| Kummen et al. [80] | 85 (55 with IBD) | PSC | Stool | 16S rRNA | *Viellonella* | Bacteria |
| Sabino et al. [81] | 52 (39 with IBD) | PSC | Stool | 16S rRNA | *Veillonella, Streptococcus, Enterococcus, Lactobacillus, Fusobacterium* | Bacteria |
| Iwasawa et al. [45] | 13 | PSC | Stool | 16S rRNA | *Veillonella, Streptococcus, Enterococcus* | Bacteria |
| Bajer et al. [82] | 43 (32 with IBD) | PSC | Stool | 16S rRNA | *Veillonella, Rothia, Streptococcus, Enterococcus* | Bacteria |
| Torres et al. [83] | 15 | PSC-IBD | Stool | 16S rRNA | *Ruminococcus, Fusobacterium* | Bacteria |
| Rühlemann et al. [84] | 73 (38 with IBD) | PSC | Stool | 16S rRNA | *Veillonella, Streptococcus, Enterococcus, Lactobacillus, Parabacterioides, Gammaproteobacteria* | Bacteria |
| Lemoinne et al. [85] | 49 (27 with IBD) | PSC | Stool | 16S rRNA | *Exophiala (fungal), Veillonella, Sphingomonadaceae, Alphaproteobacteria, Rhizobiales* | Bacteria |

ALD—alcohol liver disease; NASH—non-alcoholic steatohepatitis; NAFLD—non-alcoholic fatty liver disease; AIH—autoimmune hepatitis; PBC—primary biliary cholangitis; PSC—primary sclerosing cholangitis; IBD—inflammatory bowel disease.

### 3. Major Genetic Factors Involved in Liver Diseases Pathogenesis

With the available new technologies, studying the genetics of liver diseases has improved our understanding of them and the possibilities for therapy. These techniques include genome-wide association studies (GWAS) that allow the identification of unknown genetic risk factors, positional cloning of unknown genes associated with different diseases, the gene tests for single nucleotide variants (SNVs), and next-generation sequencing (NGS) of selected genes or/and the entire genome. Our knowledge so far confirms that gut microbiome composition and its metabolites are not only regulating factors in carcinogenesis (including de novo after liver transplantation) but also in xenobiotics and anticancer treatment failure [86–89], observations that may be related to the genetic background of the individuals.

The first GWAS reporting the most robust genetic signal for fatty liver was published in 2008 [90]. Most of the candidate genes and associations in the study have not been replicated, and their significance remains unclear. Subsequently, other GWAS studies were conducted on genetic factors for susceptibility of NAFLD, ALD, serum liver enzyme activities, hepatitis, cirrhosis, autoimmune liver diseases, etc.

This section of the paper will focus on larger GWASs conducted in recent years. These studies have described and confirmed some major genetic risk variants associated with liver disease progression, the development and severity of NAFLD and ALD, and a higher risk of cirrhosis and HCC in alcohol abusers.

The list is not exhaustive but includes identified and significant GWAS loci for predisposition and susceptibility loci for liver disease and the risk genes of progression of chronic liver disease. Multiple studies have shown that several genes, in particular, play a significant role in the pathogenesis and progression of this spectrum of diseases.

#### 3.1. GWAS Loci for Predisposition and Susceptibility of NAFLD

NAFLD is the most common form of metabolic disease worldwide, occurring in 17–30% of the population [91–93]. The etiology is thought to be multifactorial, and the heritability estimates typically range from 20 to 70%, depending on study design, methods, age, and ethnicity [94,95]. GWASs of NAFLD are relatively small because of the lack of abdominal MRI and/or liver biopsy data [90,96–104].

Several recent larger GWAS studies have been conducted that have identified multiple different risk loci for NAFLD [105–107]. Of all investigated genetic variants, patatin-like phospholipase domain-containing protein 3 (*PNPLA3*) appears to be a major common determinant of NAFLD. In 2008, the first GWAS identified a significant association between rs738409 encoding Ile148Met (I148M) and NAFLD, independent of alcohol use, body mass index, and diabetes [90]. Subsequently, multiple GWAS studies established that *PNPLA3 I148M* is strongly associated with the entire spectrum of NAFLD and genetic predisposition to disease and HCC [94,108].

Several studies have identified the *MBOAT7* variant rs641738 as a risk locus for NAFLD development and disease severity [109,110], although this variant has previously been associated with alcohol-induced cirrhosis [111].

A multi-ancestry GWAS conducted in the Million Veterans Program included 90,408 cases of chronic alanine aminotransferase elevation and 128,187 controls [112]. Seventy-seven significant genome-wide loci were identified, 25 without previous NAFLD or alanine aminotransferase associations. In two additional external NAFLD cohorts, 17 SNPs were replicated, 9 of which were novel. A pleiotropic analysis showed 61 multi-ancestry and the 17 SNPs were associated with metabolic or/and inflammatory phenotypes. Miao et al. used the UK Biobank (UKB) to estimate the NAFLD status based on anthropometric measures and serum characteristics [105]. They identified 94 NAFLD loci. Most were not previously identified but related to coronary artery disease (CAD) [105].

In 2019, Nimjou et al. published a report from a GWAS using adult and pediatric participants from the eMERGE network (Electronic Medical Records and Genomics Network) [107]. The study confirmed the association for the *PNPLA3* gene in adult and

pediatric patients, like disease severity locus. A conducted GWAS on NAFLD cases and healthy controls from the UK Biobank also identified genetic risk variants [107]. Over 9 million variants were estimated by logistic regression adjusted for sex, age, genetic components, and genotyping batch. A meta-analysis also identified six risk loci (*APOE*, *PNPLA3*, *TM6SF2*, *GCKR*, *MARC1*, and *TRIB1*). All these six susceptibility loci were significant. This GWAS also confirmed that the $\epsilon 4$ allele of *APOE* is associated with protection against NAFLD [107].

It also has been established that different cytokines are involved in the pathogenesis of NAFLD [113]. Of all, IL-6 was significantly increased in the liver of NAFLD individuals and correlated with disease severity [114]. More recently, a human pluripotent stem cell model of NAFLD was developed that is suitable for the mechanical dissection of genetic variants [115]. The study shows that the strong association between rs738409 C > G in *PNPLA3* and susceptibility to NAFLD is caused by increased IL-6/STAT3 activity, which leads to accelerated disease progression. The same study found that global blocking of IL-6 signaling reduced NAFLD development and progression. Because the IL-6 signaling pathway and the *PNPLA3I148M* variant are associated with developing HCC, this model can be used to study variants for other liver diseases than NAFLD.

### 3.2. NAFLD GWAS Loci Overlap with GWAS Loci for Liver Enzymes, ALD and HCC

Some NAFLD GWAS loci overlap with GWAS loci liver enzymes—ALT, AST, GGT, and alkaline phosphatase (ALP). They are the most commonly used laboratory markers of liver disease, and the variations in their levels are heritable [116–118]. Combined GWAS of AST and ALT have revealed genetic associations with the *PNPLA3* gene [90] and *HSD17B13* [119]. A recent study conducted a GWAS meta-analysis on serum ALT and AST activities in 411,048 subjects. They identified 100 loci associated with liver enzymes. The strongest association observed was with a rare missense variant in *SLC30A10* [120].

Another recent study on liver enzymes identified 172 *ALT*, 199 *AST*, and 216 *ALP* loci [121]. Of the mentioned loci, 160 *ALT*, 190 *AST*, and 199 *ALP* loci are novel, and 153 variants are significant. All of them are summarized and described by Chen et al., 2021. Since they are numerous and vital to liver pathology, we will focus on those overlaps and show that the liver enzyme-increasing allele increases the risk of a given disease [121]. Various studies have provided evidence for the significant heritability of alcohol dependence [122,123]. Data on the specific genetic risk variants involved in ALD pathogenesis exist. Different GWASs showed that the rs738409 variant in *PNPLA3* was associated with ALD and alcohol-related cirrhosis [124–126]. The genes *TM6SF2* and *MBOAT7* are also the genetic modifiers of ALD [111]. The MBOAT7 (rs641738) showed about 80% increased risk of HCC in NAFLD patients and the development of HCC in ALD patients [127]. The different combinations of the three variants may increase the risk of progressive liver alteration and HCC [128–131].

Stickel et al. found that the development of HCC was associated with the same genetic variants—*PNPLA3* (rs738409) and *TM6SF2* (rs58542926) [132]. Patients with alcohol-related cirrhosis who carry these variants also have an increased risk of HCC [131]. *PNPLA3*, *TM6SF2*, and *MBOAT7* appear to be genetic modifiers of both ALD and NAFLD and share some biological pathways and histological patterns [133,134].

### 3.3. GWAS Loci for Predisposition and Susceptibility of ALD

Other genetic variants involved in ALD have also been described. The studies had significantly small sample sizes, and some results were not replicated. For example, a non-synonymous variant (rs4880) in the *SOD2* gene has been associated with progressive ALD, but the data have not been confirmed [135,136].

Risk alleles in the *IL10*, *TNFα*, *TGFβ*, and *MMP-3* genes have been investigated for association with alcohol-related liver injury [137–141].

The data from these studies are not very conclusive and require further research. In addition, the rs2228603 in the *NCAN* gene is associated with NAFLD but was found to be a risk factor for HCC, even in patients affected by alcohol-related cirrhosis [142,143].

Another recent study found three loci, *ZNF827*, *GGT1*, and *HNF1A*, to be significantly associated with ALD risk [144]. Other GWASs for ALD revealed candidate genes such as *GABRB1*, *DRD4* and *TH*, *PECR*, *PDLIM5*, *METAP*, *ADH1C*, etc. [145–147].

### 3.4. GWAS Loci for Predisposition and Susceptibility of ALC

Unlike other liver diseases, very few genetic variants that influence the risk of cirrhosis have been identified. A GWAS for alcohol-related cirrhosis (ALC) in European descent identified the *MBOAT7/TMC4* locus as a new genetic risk factor [111].

Another GWAS/meta-analysis conducted in 2021 by Schwantes-An et al. analyzed ALC patients and healthy subjects who drank heavily [148]. A significant risk association was found again with *PNPLA3* and *HSD17B13*, and a protective association for *FAF2*. Meta-analysis confirmed GWAS significance for these three loci. Two other known loci, *SERPINA1* and *SUGP1/TM6SF2*, were also GWAS significant in this meta-analysis.

Emdin et al. identified 12 independent genetic variants associated with cirrhosis risk—5 previously reported and 7 newly discovered [149]. Recently identified variants include the missense variant in *APOE* (Cys130Arg) and a non-coding variant located in the 3′ untranslated region of the *EFNA1* gene (rs12904) [149].

A conducted GWAS analysis identified five previously associated variants in the *MARC1* (p.Ala165Thr), *PNPLA3* (p.Ile148Met), *TM6SF2* (p.Glu167Lys), *HSD17B13* (rs6834314), and *SERPINA1* (p.Gly366Lys) gene regions [119,150]. The previously reported variant in *MBOAT7* (rs641738) was also associated with cirrhosis. However, it did not reach genome-wide significance [103]. A recent report identified a new variant near the *HNRNPUL1* (rs15052) associated with alcoholic cirrhosis [151].

### 3.5. GWAS Loci with Significant Association with HCC

Five SNPs were found, three in *PNPLA3* and two in *SAMM50*, with significant association with NCC in conducted GWAS [152]. The SNPs in *PNPLA3* are rs2281135, rs2896019, and rs4823173. The two SNPs in *SAMM50* are rs3761472 and rs3827385. They were replicated in a cohort study in Singapore and a US case-control study, indicating that these SNPs were significantly associated with HCC. Other GWAS studies identified *WNT3A-WNT9A* (rs708113) [153]. They supported the previously reported regions associated with alcohol-related hepatocellular carcinoma risk—*TM6SF2* (rs58542926) and *PNPLA3* (rs738409) [128,154]. These two missense variations are well studied and shown to contribute to chronic liver damage by accumulating fat. Their role in liver carcinogenesis is still under investigation and remains unclear [155]. The three variants reached GWAS significance in the meta-analysis. A recent study also revealed these variants for alcohol-related HCC and described several previously reported variants [156].

As well as excessive alcohol consumption, other risk factors for HCC are chronic hepatitis B and C virus infections, obesity, aflatoxin exposure, metabolic diseases, and individual genetic predisposition. Various GWASs have been conducted for these factors, and genetic loci and their association with HCC have been well established [157]. More studies are needed for a more comprehensive understanding of the genetic mechanisms underlying alcohol-related HCC, leading to better prevention and early diagnosis.

### 3.6. Other Genetic Loci Related to Different Forms of Liver Diseases

Genetic factors related to the progression of different forms of liver diseases (ALD, NAFLD, cirrhosis, HCC, etc.) interact with genes involved in glucose and lipid metabolism, insulin signal pathways, oxidative stress, fibrogenesis, immune response, and inflammation. The liver, metabolic, and inflammatory traits are shown in Figure 4. The most significant genes associated with increased liver fat, cirrhosis, HCC, etc., are *PNPLA3*, *TM6SF2*, *HSD17B13*, *GCKR*, and *MBOAT7*. Others, such as *MARC1*, *SERPINA1*, *APOE*,

ALDH1B, GPAM, HNF1A, etc., are less stable. Rare variants such as MTTP and APOB are associated with an increased risk of liver fat damage and HCC. In addition, other genes related to the progression of NAFLD and involved in regulating lipid metabolism are *LYPLAL1*, *APOB*, *MTP*, *LPIN1*, and *UCP2*. *GCKR* has been reported to regulate glucose metabolism and lipogenesis, *IL28B* and *MERTK* in innate immunity, *SOD2* in oxidative stress, *ENPP1* and *IRS1* in insulin signaling, and *KLF6* in fibrogenesis. They are also associated with the progression of NAFLD [158–160].

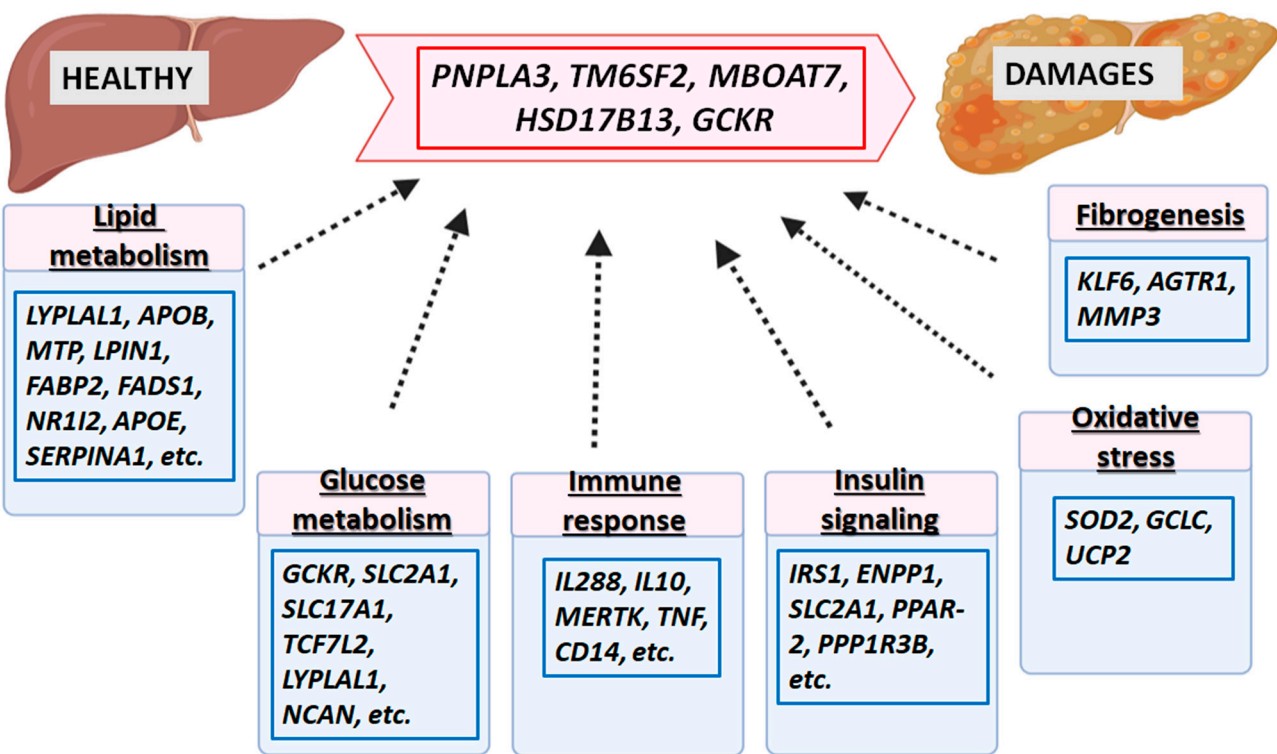

**Figure 4.** Major genetic factors involved in the pathogenesis of liver disease. The most significant genes associated with the pathology of liver diseases are given in the red box. In blue boxes are given genes involved in glucose and lipid metabolism, insulin signaling pathway, oxidative stress, fibrogenesis, immune response, and inflammation, which interact with the most important genetic factors associated with the progression of different forms of liver diseases (ALD, NAFLD, cirrhosis, HCC, etc.).

These genes also influenced microbiome composition in host genome–microbiome association studies [161]. In genetically susceptible individuals, environmental triggers may create the inability to distinguish between commensal and pathogenic microbiome components, which may cause immunological diseases. Host genetics are the leading cause of pathogenesis. The gut microbiome's role in both scenarios raises an important question: can it be used as a diagnostic biomarker or therapeutic target for autoimmune liver diseases? Except for genome editing, a person's genome is static. Probiotics, antibiotics, diet, immunization, and transplantation can alter or reassemble the human super-organism's "other genome", the microbiome.

## 4. Conclusions

Literature studies have demonstrated that the abnormalities in gut microbiota in AIH are unique to the condition. AIH and primary biliary cholangitis (PBC) can be differentiated from one another quite effectively based on the microbiota profile. Hence, the gut microbiome's functional changes in AIH can be used as non-invasive biomarkers to assess disease activity. Gut dysbiosis has been linked with the etiology of various liver diseases and different clinical profiles and biochemical responses to one-year UDCA therapy and therapeutic

failure. The scarcity of *Faecalibacterium* has been declared guilty of PBC outcome. Going further down the road, *Veillonella*'s increased abundance is registered in cirrhosis of various origins, such as AIH, PBC, or non-hepatic disorders, such as treatment-naïve Crohn's disease. The mucosal microbiota of cirrhotic patients, particularly patients with hepatic encephalopathy, differs significantly from that of healthy controls and an overgrowth of potentially pathogenic genera, both associated with poor cognition and inflammation. Moreover, cirrhotic patients have reduced conversion of primary to secondary BAs, leading to an overabundance of *Enterobacteriaceae*, *Lachnospiraceae*, *Ruminococcaceae*, and *Blautia*.

The microbiome's involvement in human health has become increasingly intriguing in recent years. The microbiome is a very intricate and heritable trait. Microbiome research in liver diseases has been developing rapidly in recent years. Studies have shown that microbial factors are critical in various liver diseases. The microbiota affects pathophysiological processes such as liver steatosis, liver inflammation, fibrosis, and hepatocellular carcinoma. More studies are needed—metagenomic, metabolomic, or even more to determine which microbiotic strains influence the phenotype of liver diseases.

Genetic, epigenetic, and environmental factors also play a decisive role in the pathogenesis and progression of liver diseases. Genetic data collected in recent years have shed light on the hereditary aspects of these diseases. But there is an urgent need to continue these studies to identify possible biomarkers for early diagnosis and personalize the treatment of higher-risk patients, as well as to study the possibilities of manipulation of the gut microbiota to be helpful in the treatment of patients with various liver diseases in early or later stages of the disease. GWASs have successfully mapped thousands of loci associated with various diseases. The method also helps scientists to identify genes related to the pathophysiology of liver and metabolic diseases and could help in the identification of new drug targets.

**Author Contributions:** Conceptualization, D.M. and T.V.; methodology, M.P.-S., V.S., M.P. and M.S.; software, G.H.V., G.V.V. and L.T.; validation, S.L., M.G. and T.V.; formal analysis, D.M. and A.M.; investigation, M.P.-S., V.S., M.P. and M.S.; resources, G.H.V. and L.T.; data curation, S.L. and M.G.; writing—original draft preparation, D.M., M.P.-S. and T.V.; writing—review and editing, T.V.; visualization, D.M.; supervision, T.V.; project administration, T.V.; funding acquisition, T.V. All authors have read and agreed to the published version of the manuscript.

**Funding:** This study is financed by the European Union-NextGenerationEU, through the National Recovery and Resilience Plan of the Republic of Bulgaria, project № BG-RRP-2.004-0008-C01.

**Institutional Review Board Statement:** Not applicable.

**Informed Consent Statement:** Not applicable.

**Data Availability Statement:** Not applicable.

**Acknowledgments:** This study is financed by the European Union-NextGenerationEU through the National Recovery and Resilience Plan of the Republic of Bulgaria, project № BG-RRP-2.004-0008-C01.

**Conflicts of Interest:** The authors declare no conflict of interest.

## Abbreviations

| | |
|---|---|
| NAFLD | Non-alcoholic fatty liver disease |
| ALD | Alcohol-related liver disease |
| AIH | Autoimmune hepatitis |
| PSC | Primary sclerosing cholangitis |
| PBC | Primary biliary cholangitis |
| HCC | Hepatocellular carcinoma |
| GWASs | Genome-wide association studies |

|  |  |
|---|---|
| SNVs | Gene tests for single nucleotide variations |
| NGS | Next-generation sequencing |
| TMA | Trimethylamine |
| TMAO | Trimethylamine N-oxide |
| MAMPs/PAMPs | Microbial-(or pathogens-) associated molecular patterns |
| BAs | Bile acids |
| VLDL | Very low-density lipoprotein |
| FMT | Fecal microbiota transplantation |
| AST | Aspartate transferase |
| GGT | Gamma-glutamyl transpeptidase |
| APL | Alkaline phosphatase |
| ALT | Alanine transaminase |
| LPS | Lipopolysaccharide |
| RCT | Randomized clinical trial |
| AhR | Aryl hydrocarbon receptor |
| SLA/LP | soluble liver antigen/liver–pancreas |
| anti-SLA/LP | Anti-soluble liver antigen/liver–pancreas |
| UDCA | Ursodeoxycholic acid |
| SCFAs | Short-chain fatty acids |
| IBD | Inflammatory bowel disease |
| UKB | United Kingdom Biobank |
| CAD | coronary artery disease |
| LPS | Lipopolysaccharide |
| eMERGE network | Electronic Medical Records and Genomics Network |

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
