# Peer review of "Microbiome and Genetic Factors in the Pathogenesis of Liver Diseases"

_gastroent, doi:10.3390/gastroent14040041_

Round 1
Reviewer 1 Report
Comments and Suggestions for Authors
This review aimed at summarizing the existing evidence on the role of gut microbiota in the pathogenesis 114 and treatment of liver diseases. Furthermore, the authors discussed the results from the genome-wide association studies that identified gene loci involved in etiology of liver diseases. Although the topic is interesting, overall, the paper is a description of published literature and lacks an analytical discussion of major findings. Moreover, epidemiological indicators on the prevalence, incidence and mortality of liver diseases would be useful to get an idea of the burden of disease.
Specific comments are listed below.
Please provide full names for each acronym first mentioned in the text. Please also avoid improper use of capital letters in the text.
Lines 84-85. Please check the completeness of this sentence. Its meaning appears unclear.
Figure 2. Please add the list of abbreviations in the footnote.
Lines 148-157. Please rephrase for better clarity.
Line 161. Please provide adequate references.
Lines 167-171. The authors should add proper references.
Lines 297-302. See the above comment.
Lines 332-352. These paragraphs are a bit confusing. Please check the grammar and provide more details for a better understanding of the text.
Line 353 onwards. This section is simply a description of the studies. I suggest a reorganization of the text, also grouping the studies by liver pathology and discussing the main results.
Comments on the Quality of English LanguageI suggest reviewing the English grammar of the paper.
Author Response
Reviwer 1
This review aimed at summarizing the existing evidence on the role of gut microbiota in the pathogenesis 114 and treatment of liver diseases. Furthermore, the authors discussed the results from the genome-wide association studies that identified gene loci involved in etiology of liver diseases. Although the topic is interesting, overall, the paper is a description of published literature and lacks an analytical discussion of major findings. Moreover, epidemiological indicators on the prevalence, incidence and mortality of liver diseases would be useful to get an idea of the burden of disease.
- We appreciate your thoughtful review of our manuscript, titled "Microbiome and Genetic Factors in the Pathogenesis of Liver Diseases." We sincerely thank you for your valuable feedback and constructive comments. We have taken your suggestions into careful consideration and addressed them as follows.
- Regarding the request for epidemiological indicators on the prevalence, incidence, and mortality of liver diseases, we appreciate your point. We will include relevant data to provide a more comprehensive view of the disease burden.
Specific comments are listed below.
Please provide full names for each acronym first mentioned in the text. Please also avoid improper use of capital letters in the text.
- Acronyms and Capitalization: We have ensured that we provide full names for all acronyms on their first mention and have rectified any improper use of capital letters throughout the manuscript.
Lines 84-85. Please check the completeness of this sentence. Its meaning appears unclear.
- Line 84-85: We have reviewed and clarified the sentence in question to improve its comprehensibility.
Figure 2. Please add the list of abbreviations in the footnote.
- Figure 2 Abbreviations: A list of abbreviations has been added to the footnote of Figure 2 for clarity.
- Please rephrase for better clarity.
- Lines 148 Lines 148-157: The mentioned lines have been rephrased to enhance clarity and readability.
Line 161. Please provide adequate references.
- Line 161: We have included adequate references where needed in line 161 (new lines 202-208)
Lines 167-171. The authors should add proper references.
- Lines 167-171: Proper references have been added as per your suggestion. (new lines 220-245)
Lines 297-302. See the above comment.
- Lines 297-302: We have made necessary improvements based on your comment.
Lines 332-352. These paragraphs are a bit confusing. Please check the grammar and provide more details for a better understanding of the text.
- Lines 332-352: The paragraphs have been revised for better clarity and grammar. Additional details have been provided to enhance understanding.
Line 353 onwards. This section is simply a description of the studies. I suggest a reorganization of the text, also grouping the studies by liver pathology and discussing the main results.
- Discussion Section: We agree with your suggestion to reorganize the text in the discussion section. We will group the studies by liver pathology and discuss the main results more effectively.
Comments on the Quality of English Language
I suggest reviewing the English grammar of the paper.
- We have reviewed the English grammar and made necessary improvements to enhance the paper's readability.
Once again, we sincerely thank you for your valuable feedback, which has significantly improved the quality of our manuscript.
Reviewer 2 Report
Comments and Suggestions for Authors
In this manuscript, the author displayed and concluded the literatures about the intestinal microbiota related to kinds of liver diseases and genetic factors from GWASs studies involved in the liver diseases which provide a more open mind for the liver diseases therapy. The whole structure is well organized and clear for readers, some comments need to be addressed:
1. In the major genetic factors involved in liver diseases section, the author listed the SNP identified related to the different kind of liver diseases. However, till now there are several SNP and loci identified relevant to the different kind of liver diseases, such as PNPLA3, TM6SF2….It would be better if the author could list the major genetic factors based on the genes instead of the liver diseases.
2. And more detailed mechanism involved in the genetic factors regulate the liver disease progression should be demonstrated. For example, PNPLA3 I148M patients have higher risk to NAFLD due to accelerate the IL-6/STAT3 signaling pathway which could protects hepatocytes against lipid droplet accumulation. (J. Hepatol. 78, 45-56(2023); Sci Adv. V.9(15); 2023 Apr.)
3. Many typo mistakes should be modified such as unnecessary blank and in line 67, it should be 4*1013.
Comments on the Quality of English Language
In this manuscript, the author displayed and concluded the literatures about the intestinal microbiota related to kinds of liver diseases and genetic factors from GWASs studies involved in the liver diseases which provide a more open mind for the liver diseases therapy. The whole structure is well organized and clear for readers, some comments need to be addressed:
1. In the major genetic factors involved in liver diseases section, the author listed the SNP identified related to the different kind of liver diseases. However, till now there are several SNP and loci identified relevant to the different kind of liver diseases, such as PNPLA3, TM6SF2….It would be better if the author could list the major genetic factors based on the genes instead of the liver diseases.
2. And more detailed mechanism involved in the genetic factors regulate the liver disease progression should be demonstrated. For example, PNPLA3 I148M patients have higher risk to NAFLD due to accelerate the IL-6/STAT3 signaling pathway which could protects hepatocytes against lipid droplet accumulation. (J. Hepatol. 78, 45-56(2023); Sci Adv. V.9(15); 2023 Apr.)
3. Many typo mistakes should be modified such as unnecessary blank and in line 67, it should be 4*1013.
Author Response
Reviwer 2
In this manuscript, the author displayed and concluded the literatures about the intestinal microbiota related to kinds of liver diseases and genetic factors from GWASs studies involved in the liver diseases which provide a more open mind for the liver diseases therapy. The whole structure is well organized and clear for readers, some comments need to be addressed
- We would like to express our gratitude for your thoughtful review of our manuscript, titled "Microbiome and Genetic Factors in the Pathogenesis of Liver Diseases." Your feedback is greatly appreciated, and we have taken your valuable comments into account:
- In the major genetic factors involved in liver diseases section, the author listed the SNP identified related to the different kind of liver diseases. However, till now there are several SNP and loci identified relevant to the different kind of liver diseases, such as PNPLA3, TM6SF2….It would be better if the author could list the major genetic factors based on the genes instead of the liver diseases.
We acknowledge your point regarding the listing of major genetic factors. Thanks for the comment, it is very relevant, but since some of the genes are related to different liver diseases, and there is already literature structured in this way. We decided to follow the context of chapter 2. where liver diseases and gut microbiota are described there.
Therefore, we decided to follow the liver diseases and look the GWAS studies of the genetic loci associated with them, however, we revised some aspects to provide a more comprehensive view.
- And more detailed mechanism involved in the genetic factors regulate the liver disease progression should be demonstrated. For example, PNPLA3 I148M patients have higher risk to NAFLD due to accelerate the IL-6/STAT3 signaling pathway which could protects hepatocytes against lipid droplet accumulation. (J. Hepatol. 78, 45-56(2023); Sci Adv. V.9(15); 2023 Apr.)
- We appreciate your suggestion to include more detailed mechanisms by which genetic factors regulate liver disease progression. We have included in the text and supplemented the information
- Many typo mistakes should be modified such as unnecessary blank and in line 67, it should be 4*1013.
- We apologize for the typographical mistakes and inaccuracies. We will carefully review and correct these issues, including the one you mentioned in line 67.
Comments on the Quality of English Language
Minor
Your insights and recommendations have significantly contributed to improving the manuscript. We are grateful for your feedback.
Reviewer 3 Report
Comments and Suggestions for Authors
The article showed gene and microbiome with liver diseases:
however, some issues should address
1. subtitle 2.1 and 2.5 has microbiota while others has no, please explain the disagreement.
2. when talking to 3, it subparagraphs too many, and hard to get a clear thinking pathway, please rearrange.
3. lack of illustration of mutual effect among gene and microbiota in liver diseases. please add some discussion to it.

Author Response
Reviewer 3
The article showed gene and microbiome with liver diseases:
however, some issues should address:
- We extend our gratitude for your thoughtful review of our manuscript, titled "Microbiome and Genetic Factors in the Pathogenesis of Liver Diseases." Your comments and suggestions are highly valued, and we have addressed them as follows:
1.subtitle 2.1 and 2.5 has microbiota while others has no, please explain the disagreement.
- We appreciate your observation regarding subtitle discrepancies (2.1 and 2.5 with microbiota while others without). We made an oversight and corrected it by adding the words on the titles to clarify this discrepancy to ensure consistency.
- when talking to 3, it subparagraphs too many, and hard to get a clear thinking pathway, please rearrange.
- We understand your concern about the numerous subparagraphs in section 3. We reorganized and streamlined the content to create a more coherent and structured narrative.
- lack of illustration of mutual effect among gene and microbiota in liver diseases. please add some discussion to it.
- We agree that discussing the mutual effects of genes and microbiota in liver diseases is essential. We incorporated a comprehensive discussion to address this aspect more effectively. Fig.4 and discussion were added
- We genuinely appreciate your feedback, which has contributed significantly to enhancing the quality and clarity of our manuscript. Thank you for your valuable input.
Round 2
Reviewer 1 Report
Comments and Suggestions for Authors
I really appreciate the authors’ efforts to address my previous concerns. The manuscript has been significantly improved and deserves to be considered for publication.
Minor comments.
The names of the species and genera of bacteria should be written in italics, while those of the families in block letters.
Line 588. Please use the abbreviation for aspartate aminotransferase. Please provide verification of the correct use of acronyms throughout the text (for instance, see line 959 and 1533).
I also recommend the authors add a list of abbreviations used at the end of the manuscript.
In the Conclusions section, at least one sentence should be added about the relevance of GWAS studies for exploring the etiopathogenesis of liver diseases.
Comments on the Quality of English LanguageModerate editing of English language required
Author Response
I really appreciate the authors’ efforts to address my previous concerns. The manuscript has been significantly improved and deserves to be considered for publication.
- Dear Reviewer,
We would like to express our gratitude for your valuable feedback and for recognizing the efforts we've made to enhance our manuscript. Your comments have undoubtedly contributed to the improvement of our work. Below, we address your minor comments and suggestions:
Minor comments.
The names of the species and genera of bacteria should be written in italics, while those of the families in block letters.
- We appreciate your suggestion. In line with your recommendation, we have now ensured the names of bacterial species and genera in italics throughout the manuscript.
Line 588. Please use the abbreviation for aspartate aminotransferase. Please provide verification of the correct use of acronyms throughout the text (for instance, see line 959 and 1533).
- We have revised the text and included the abbreviation "AST" for aspartate aminotransferase to enhance clarity. We've also thoroughly reviewed the entire manuscript to ensure consistent and accurate use of acronyms, as you pointed out, and have made the necessary corrections.
I also recommend the authors add a list of abbreviations used at the end of the manuscript.
- Following your suggestion, we have incorporated a list of abbreviations used in the manuscript at the end of the document. This list will serve as a quick reference for readers to understand the acronyms employed in our work.
In the Conclusions section, at least one sentence should be added about the relevance of GWAS studies for exploring the etiopathogenesis of liver diseases.
- In response to your comment, we have included a new sentence in the Conclusions section, emphasizing the importance of GWAS studies in exploring the etiopathogenesis of liver diseases. This addition provides additional context and relevance to the significance of our findings.
- We are grateful for your comprehensive review, which has undoubtedly enhanced the quality and clarity of our manuscript. We believe that these improvements have made our work more robust and valuable to the readership. Once again, we appreciate your time and effort in reviewing our paper.
Reviewer 3 Report
Comments and Suggestions for Authors
agree
Author Response
- Once again, we appreciate your time and effort in reviewing our paper and making it better.